# The effect of tempi and mode on the rating of the perceived emotion in music

Ulvhild Helena Færøvik[1]*, Karsten Specht[1,2]

**1** Department of Clinical and Biological Psychology, Faculty of Psychology, University of Bergen, Bergen, Norway, **2** Department of Radiology, Mohn Medical Imaging and Visualization Centre, Haukeland University Hospital, Bergen, Norway

* Ulvhild.eide@uib.no

## Abstract

In this study, we investigated the impact of tempi variations (60, 100, 120, and 150 beats per minute) and major and minor modes on the perceived emotional content of music. To explore this, we created eight versions of five original compositions, resulting in 40 musical stimuli. Control stimuli included variants of white, brown, and pink noise, as well as six human voice recordings. A total of 1280 participants took part in an online survey. Participants were diverse, comprising 262 musicians (defined as individuals with over six years of instrument-playing experience who identified as amateur, semi-professional, or professional). Participants were asked to rate the perceived emotional content of the stimuli, using the second-order factors of the GEMS-9 framework: 'sublimity', 'unease', and 'vitality'. We employed a mixed between-within-subjects analysis of variance to analyse the data. Specifically examining the influence of the four tempi, the two modes, while controlling for musicianship, gender, age, and education level. Results showed that as tempi increased, compositions were consistently rated as less sublimity and more vitality, in both major and minor modes. A higher tempi in the major mode resulted in lower ratings of uneasiness. There were also several significant findings for musicianship, gender, and education level, although these mostly had small effect sizes. For age groups, there were two substantial differences between the groups with larger effect sizes. Overall, our findings suggest that tempi and mode significantly influence how compositions are rated for sublimity, unease, and vitality, as they interact.

## Introduction

Listening to music is a ubiquitous and inherently emotional experience, shaped by a myriad of factors. We wanted to examine tempi and mode while controlling for musicianship, gender, age, and educational level. Yet, many theories exist about measuring and conceptualising emotion in music. Despite research on the impact of musical

**Data availability statement:** Data is uploaded to figshare: https://doi.org/10.6084/m9.figshare.30052507.

**Funding:** This project was financed through a grant from the Research Council of Norway (Grant Number: 217932/F20) awarded to KS. The salary of UF was covered in part by a master's student scholarship at the University of Bergen, and in part by a grant from the Research Council of Norway (Grant Number: 260576) awarded to Stefan Koelsch. The funders had no role in study design, data collection and analysis, decision to publish, or preparation of the manuscript.

**Competing interests:** The authors have declared that no competing interests exist.

structures, key constructs like tempi and mode are examined using heterogeneous emotion models, stimuli and analytical frameworks [1].

In the following, we briefly introduce the main approaches used to conceptualise and measure emotions in music, to motivate the choice of emotion model used in the present study. Additionally, we will also give a brief overview of previous research on emotional perception of tempi, mode in music, and some covariates like age, gender, and musicianship.

Research on music and emotion typically relies on discrete, dimensional, or domain-specific emotion models, which differ in their theoretical assumptions and measurement focus. It is essential to distinguish between emotion theories (emotivist, cognitivist, and constructionist, emotion measurement models (discrete and dimensional) and phenomenological target of measurement (felt and perceived emotion).

Discrete emotion models are rooted in basic emotion theory and assume that a limited set of universal emotions can be evoked or perceived through music [2,3]. These models typically use categorical emotion labels such as happiness, sadness, anger, and fear [4] and assume that music evokes emotions in a manner comparable to other affective stimuli, such as Ekman's facial expressions [5].

Discrete emotion approaches are prevalent in music and emotion research because basic emotions are considered relatively universal and listeners tend to identify them reliably in musical stimuli [6,7]. However, a central limitation of basic emotion theory is the lack of consensus regarding which emotions should be considered "basic" [8]. As a measurement framework, discrete emotion models can be applied to both felt and perceived emotions, as the felt–perceived distinction primarily concerns task instructions rather than the underlying emotion model itself [4].

An alternative approach is the dimensional model of emotion, often grounded in appraisal and constructionist theories [9]. Emotions are considered continuous along dimensions such as valence and arousal and can be applied to both felt and perceived experiences [2]. In this context, listeners typically rate music in terms of valence, arousal, or related constructs such as tension [10]. For example, both scary and happy music are often associated with high arousal, whereas pleasant and sad music are associated with low arousal [11]. While dimensional models allow for graded and continuous emotion ratings, they may provide a relatively coarse description of emotional experience and may not fully capture the aesthetic qualities of musical emotions.

Studying emotions in music presents additional challenges, as emotional responses to music are often less intense than those elicited by life events or autobiographical memories [12]. Scherer, therefore, argues that music should be studied as a conscious feeling that can be measured cognitively and physiologically [13]. This perspective is reflected in the component process model of emotion (CPM), which explains emotional induction and perception in music based on musical structure, stimulus characteristics, and listeners' subjective and physiological responses [14,15]. Similarly, it has been shown that regardless of basic or dimensional models, emotions in music can be both evoked and perceived with comparable emotional qualities [16]. In line with this dynamic view of emotion, the construction organisation

dynamics appraisal (CODA) model emphasises that emotions are constructed from basic psychological components and evolve through continuous appraisal and reappraisal during music listening [17].

In response to the limitations of general emotion models, domain-specific emotion models have been developed to capture emotions that are particularly relevant to music. A literature review of music and emotion research indicates that only a minority of studies employ music-specific emotion models [18]. One model is the Geneva Emotional Music Scale-9 (GEMS-9) [19], which was specifically developed to assess aesthetic emotions in music. The GEMS-9 consists of nine first-order factors: wonder, transcendence, tenderness, nostalgia, peacefulness, joyful activation, power, tension, and sadness. Each first-order factor represents multiple emotional words. The GEMS-9 can also be collapsed into three meta-factors, or second-order factors: sublimity, vitality, and unease [19]. Importantly, the GEMS-9 was developed to test both felt and perceived emotions in music.

Although the distinction between felt and perceived emotions remains debated, most music stimuli are used to assess perceived rather than felt emotions [4]. By focusing on aesthetic emotions that are characteristic of musical experiences, the GEMS-9 provides a music-specific framework for studying emotional responses to music that are less directly tied to utilitarian action tendencies [20].

One strength of the GEMS-9 is that it has been validated across a wide range of music genres, from classical to techno. The GEMS can also generalise well to music styles beyond those included in its original development, such as neoclassical, ambient, new age, punk, folk rock, electronica, electro-medieval, disco, heavy metal, and opera [21–23]. This broad applicability supports the use of the GEMS-9 as a domain-specific emotion model capturing aesthetic emotions across diverse musical contexts.

At the same time, the GEMS-9 has been described as less efficient than discrete and dimensional emotion models [24,25]. Despite this lower efficiency, the GEMS-9 contains both the strongest and weakest items when compared to discrete and dimensional models, indicating substantial variability in item-level performance [24,25]. In contrast, the authors of the GEMS-9 model reported that GEMS-9 outperformed discrete and dimensional models [19]. Together, these findings suggest that while the GEMS-9 may be more complex than general emotion models, it offers advantages in sensitivity and domain relevance.

The current study was part of a larger project on music and emotion, which aimed to measure both the emotional induction, through physiological data, and the perception of emotion through self-report ratings. Hence, we chose to use the thoroughly tested GEMS-9 scale, which was developed to measure both evoked and perceived emotion [19]. To balance domain specificity with measurement efficiency, we focused on the second-order structure of the GEMS-9. Halpern proposed that the GEMS could be condensed into three or four overarching categories, effectively capturing higher-level dimensions of musical emotion while retaining the model's aesthetic focus [26]. Accordingly, the GEMS-9 can be collapsed into three second-order factors—sublimity, vitality, and unease, which summarise the covariance among the nine first-order factors [19]. Specifically, sublimity shows strong correlations with wonder (1.00), tenderness (0.75), nostalgia (0.89), transcendence (0.66), and peacefulness (0.65). Vitality is defined by near-perfect correlation with power (1.00) and joyful activation (0.95), indicating high internal reliability. Unease has a perfect correlation with tension (1.00), but not with sadness (0.27). Reflecting both a dominant core component and meaningful differentiation within the factor. Together, these strong within-factor associations and the clear separation between second-order factors support collapsing the nine first-order GEMS dimensions into three reliable and interpretable higher-order factors [19,26].

Using the second-order factors allows for a more parsimonious representation of emotional responses to music while preserving the domain-specific conceptualisation of aesthetic emotions embedded in the GEMS framework. Although these second-order factors resemble dimensional emotion spaces, they differ from generic valence–arousal models by reflecting emotion categories derived specifically from musical experience rather than from general affective theory. Thus, the use of GEMS-9 second-order factors provides a theoretically grounded and psychometrically supported compromise between expressive richness and analytical efficiency. For these reasons, the GEMS-9 second-order factors were selected as the emotion model in the present study.

Tempi, defined as the frequency rates of underlying pulse structures [27], is commonly expressed in beats per minute (BPM) [28]. Fast tempi are associated with activation, happiness, tension, arousal, anger, and fear [29–32]. Whereas slow tempi tend to evoke emotions of calmness, peace, sadness, tenderness, longing, boredom, and disgust [29,32]. Further, higher tempi and valence ratings are associated with higher arousal [33].

Early research [34] postulates that elements like tempi and rhythm profoundly influence bodily movement and motor response, supporting the idea that motion in music plays a significant role in emotional expression. This is corroborated in another study [28], showing that small children master tempi perception before mode perception. Similarly, patients with cochlear implants also rely on tempi for emotional ratings when listening with implant-ear alone [35]. Another study found that arousal in music was determined by tempi, while valence was determined by mode [36].

The intricate relationship between tempi and emotional states not only highlights its role as a fundamental aspect of music expression but also underscores its influence on our physiological and emotional responses.

Major and minor triads (not being named thus) became the most general trichords in Renaissance music [37]. In Western music, emotional differences between major and minor tonalities might have emerged at the same time, between the 14th and 17th centuries, when the tonal system emerged [38]. A recent systematic review posits that the major-minor dichotomy represents the primary emotional feature in Western music [39].

A musical key is defined by tonic and scale, commonly referred to as mode, typically categorised as either minor or major [40]. Mode can affect valence and arousal ratings in music [41,42]. Major modes typically convey a sense of happiness, while minor modes are linked to feelings of sadness [32]. Despite minor modes being linked with negative valence, this association is modulated by the tempi at which music is played [34]. Such nuances emphasise the complex interplay of musical elements in shaping emotional experiences for listeners.

For example, listeners understand emotions in Hindustani ragas, even though they are unfamiliar with the tonal system [43]. The authors report that listeners relied on tempi and mode to perceive joy, sadness, and anger. Further, Chinese students rated music in a major key as higher on pleasure, arousal, and dominance compared to a minor key [44]. Interestingly, regardless of mode, music in slow tempi has been rated shorter than music in moderate and fast tempi, suggesting that tempi affects the subjective time perception [45].

Another example is the study by Chubb and coworkers (2013) across three experiments on tone scrambles in either major or minor across 275 listeners. They found that listeners tend to be sorted into two subpopulations. One group is sensitive to major and minor, and the second group is not. Years of music training showed only a modest correlation of around 0.5, and thus a preexisting listening sensitivity explains why some are better at distinguishing major from minor [46].

Previous research has not found a difference in the perception of emotion in music regarding gender [47], and gender and expertise [48]. However, children perceived more happiness and less anger in music compared to adults [48]. Therefore, it is important to control for gender, age, education level, and expertise when doing studies on the perception of music. A systematic review found that perception of major and minor modes is influenced by age, culture, personality and health [49].

Building on age differences, they have been noted in several studies on music perception [25,50,51]. Interestingly, older participants reported stronger emotions toward happy music and less sadness and scary music [52]. Furthermore, adolescents and older adults tend to perceive music more positively than young and middle-aged adults [50]. However, it has been shown that middle-aged adults rate sadness and fear in music lower than young adults [53]. Taken together, these studies show quite varying results for age differences and music perception.

Another important feature of our current study was to have pilot-tested control stimuli. Some argue for acoustical control stimuli, emphasising their importance in contrast with music conditions [54]. Others posit that control stimuli should be non-musical, highlighting the inconsistency of the optimal choice of control conditions in music-related research [55].

It is also common to compare music to control conditions, such as silence and unfamiliar music [56]. However, silence as a control condition could increase the potential for a placebo effect, particularly when other conditions have an auditory

component [57]. In two studies, the authors discovered the importance of an active control condition. They observed an effect of music in the first study (silence as control) but not in the second (radio show as control), leading to a discussion on the impact of active versus passive control conditions [58].

A passive control stimulus is something that does not require much attention from the participant, such as silence. An active control stimulus is something that requires attention from the participant, such as giving a subjective or physiological response. While researchers often try to use a "neutral" stimulus as a control condition, the stimuli have typically not been validated beforehand, which may lead to biases and false conclusions [59]. As such, we opted for various control stimuli, using both voice and noise controls.

To contribute to a more nuanced understanding of the perception of emotion in music, our study aimed to explore how alterations in tempi and mode influence ratings, using multiple control stimuli and a large data sample. We used self-composed music, unknown to the participants, since familiarity with music can confound the results of studies [4]. Additionally, we sought to control the impact of musicianship, gender, age, and education level on ratings of perceived emotion.

We acknowledge that there has been a lot of research on the effect of tempi and mode on the perception of music and emotion. While tempi and mode effects are established, their robustness across large samples and within a music-specific aesthetic framework remains underexplored.

Although the effects of tempi and mode on emotional ratings are well established, much of the existing literature relies on small samples, heterogeneous stimuli, and varying emotion models. High-powered conceptual replications using controlled, unfamiliar musical material remain comparatively rare. Without going into the discussion on the replication crisis, it is crucial to note the importance of robustness and generalisability. Particularly in domains where effect sizes are often small, and stimuli vary widely. By combining a large sample, original and pilot-tested compositions, and multiple control stimuli, the present study aims to provide a rigorous confirmatory test of tempi and mode effects within a music-specific emotion framework. As such, the current study is not a replication study, but a conceptual replication study.

## Hypothesis

Building upon prior research, our four-point hypothesis is as follows:

H1. Tempo will positively predict vitality across both modes.

H2. Mode will predict unease independent of tempo.

H3. Tempo and mode will interact such that tempo attenuates minor–unease effects.

H4. There will be differences in emotional ratings based on gender, age, musicianship, and education.

## Method

### Stimuli

Five compositions were chosen from a selection of original pieces. We edited each composition in Logic Pro X [60] to adjust the tempi to 60, 100, 120, and 150 beats per minute (BPM). Each composition lasted 21 seconds. The stimuli were composed with simple melodies, without melodic changes, harmonic tirades, or elements of surprise that could alter the perceived tempi [61]. We also checked that the tempi beat in the compositions were perceived correctly in a small, unpublished pilot study (N = 82). We specifically composed the stimuli to control the musical parameters of tempi and mode. In a previous but similar pilot study, we also investigated dynamics in addition to tempi and mode, and we found that tempi and mode were the parameters that influenced emotional ratings the most, available as a preprint [62].

Additionally, we altered the compositions into a major mode and a minor mode. Consequently, each of the five compositions was played once in every tempi, in both major and minor modes, resulting in each composition being played a total of eight times across all conditions. The survey encompassed 40 compositions.

Three additional original compositions had been previously pilot-tested and available as a preprint [62] and identified as exemplary baselines for sublimity, unease, and vitality compositions. These three compositions served as test stimuli and were presented at the outset of the survey.

To control for the influence of music stimuli versus other auditory stimuli in the study, three types of noise were chosen: variants of white, pink, and brown noise. Additionally, neutral voice recordings with one female and one male speaker, each reading three different texts in an eastern Norwegian dialect, were selected. This selection resulted in a total of six piloted audio recordings and three additional noises. All music stimuli and control stimuli can be listened to here (S1 File: https://figshare.com/projects/Music_and_control_stimuli/246239). See sheet music in S1 File.

## Materials

The studies were conducted using an internet survey based on l.a.m.p. (Linux/Apache/MySQL/PhP) and HTML/Javascript/CSS, which was developed in house.

The second-order factors from the GEMS-9 were used to assess the emotional validation of all stimuli [19]. This modification, aimed to align the GEMS model more closely with a dimensional approach while still ensuring comprehensive testing within the domain of music and emotion scales [26]. We translated the GEMS items test into Norwegian at our department in 2014. Two students (including the first author) translated the words to English, and a third student back translated to English. Detailed description of the words is available in the supporting information (S1 File). By using three second-order factors, we also counterbalanced the GEMS-9, as the first-order factors consist of five emotion words for sublimity (overweight) but only two for unease and vitality, respectively. This makes the GEMS-9 more scalable for participants to be able to choose three items that do not overlap, compared to five categories that do overlap (sublimity first-order factors).

To aid participants in understanding the emotional terms, a comprehensive description of the first-order factors from the GEMS-9, as well as additional synonyms, was presented alongside the sounds. See supporting information for the Norwegian synonyms we used, with English translation (S1 File).

Additionally, for the demographic section of the survey, a subset of questionnaire items from the Profile of Music Perception Skills (PROMS) [63] was incorporated. The PROMS is a questionnaire that measures the musical abilities of musicians and non-musicians.

## Participants

The initial sample comprised 1280 individuals, 757 of whom identified as female, 496 as male, and 27 who chose not to disclose their gender. The participants' ages ranged from 19 to 94 (M = 40, SD = 15.6). To be included in the data analysis, participants were required to answer more than 70% of the survey.

Regarding educational attainment, one participant had received no formal education, 22 had basic education, 218 had completed high school, 573 had higher education qualifications, 442 had a master's degree or PhD as the highest completed level of education, and 24 participants chose not to provide their educational background.

Musicians were operationally defined as individuals who had played a musical instrument for over six years and identified as amateur, semi-professional, or professional musicians (N = 262).

## Power

The target for respondents was set at 1562 to have a representative sample for the Norwegian population of 5 million people. We calculated the number with a sample size calculator (https://www.calculator.net/sample-size-calculator.html) with a

95% confidence level and a 2.48 confidence interval. With our obtained sample size of 1280 participants, this corresponds to a margin of error of ±2.68% at a 95% confidence level. Using G*Power [64,65], Post hoc power analysis indicated statistical power approaching 1.00, with a critical F value of 11.85, when the effect size was set to.2, for the sample of 1280.

## Ethics

Participants were recruited through snowball sampling [66] via social media. They were eligible to join if they were over 18 and had normal or corrected hearing. This information was screened before analysis. Recruitment for the survey started on June 1, 2020, and ended on December 1, 2022.

All procedures were approved by the Regional Committee for Medical and Health Research Ethics (REK 45655) and carried out per the Code of Ethics of the World Medical Association, Declaration of Helsinki. Upon agreeing to participate, they received a username, a password, and a link to access the online survey. All participants gave electronic informed consent as part of the initial page of the survey before they could access the rest of the survey.

## Procedure

Participants received a link, individual passwords, and usernames to access the survey. They were explicitly instructed to wear headphones, although this could not be verified or tested by us.

The survey, encompassing the full set of stimuli, took approximately 21 minutes to complete if participants listened to the full 21 seconds of each stimulus. However, the average completion time among participants was 16 minutes and 44 seconds, indicating that most participants spent less than 21 seconds listening to the stimulus. We excluded the times of 17 participants that exceeded 2 hours for the mean calculation of completion time. This exclusion was made under the assumption that extended times may be attributed to participants leaving the survey tab open on their computers.

The opening section of the survey requested demographic information from participants, including gender, age, education level, handedness, and details about music training. The initial part of the survey presented three examples of compositions that had undergone pilot testing and were identified as sublimity, vitality, and unease, available as a preprint [62].

After completing the test page, participants progressed to the main survey section, where one stimulus was presented at a time.

Participants were instructed to rate all compositions, noises, and voice recordings using the second-order factors from the GEMS-9 questionnaire. Each composition was to be rated on a seven-point Likert scale, ranging from 1 (indicating very little) to 7 (indicating very much). For each sound, participants were asked to provide ratings of perceived sublimity, unease, and vitality.

Participants had the flexibility to advance to the next sound as soon as they had rated the stimuli on all three emotions, allowing them the option to not necessarily listen to the full 21 seconds of the stimuli. The survey design also allowed participants the freedom to play each stimulus multiple times if desired. To maintain control and consistency, every fourth stimulus was a control sound, noise, or voice. The presentation order of stimuli was semi-randomised.

## Analysis

Participants who completed less than 70% of the survey were excluded from the respective analyses. Because missing responses varied slightly across conditions, the effective sample size differs marginally between models.

The primary aim of the analysis was to examine the effects of tempo and mode, rather than differences between individual compositions. Each participant rated five structurally comparable compositions under each tempo–mode condition. To obtain robust condition-level estimates and reduce stimulus-specific variance, ratings were aggregated across the five compositions within each tempo–mode combination.

The aggregation procedure was conducted in the following steps:

For each participant, emotion ratings (sublimity, unease, vitality) were recorded separately for each composition. Within each tempo–mode condition (e.g., 60 BPM major), ratings were averaged across the five compositions.

This resulted in eight condition-level variables per emotion (4 tempi × 2 modes).

Before aggregation, descriptive statistics were inspected at the composition level. Mean ratings and variability were highly similar across the five compositions within each tempo–mode condition (S1 File), supporting aggregation. This procedure reduces idiosyncratic stimulus effects and strengthens inference at the parameter level (tempo and mode).

Control stimuli (voice and noise) were analysed separately and were not included in the factorial tempo–mode models.

**Statistical model.** Data analysis was conducted in IBM SPSS Statistics (version 29.0.20). To examine the effects of tempo and mode on emotional ratings, we conducted mixed-design analyses of variance (ANOVAs) separately for sublimity, unease, and vitality.

The design included: Within-subject factors with tempi (60, 100, 120, 150 BPM) and mode (major, minor). Between-subject factors: Musicianship (musician vs non-musician) and Gender. Covariates: Age (continuous) and Education level.

The full factorial model tested main effects of tempo and mode, as well as their interaction, while controlling for demographic variables and their interactions with the within-subject factors.

Using Wilkinson–Rogers notation [67], the tested model can be expressed as:

Emotional rating ~ Tempo × Mode + Musicianship + Gender + Age + Education

Separate models were estimated for each emotional dimension.

To control the overall type 1 error rate (α), the Bonferroni-type approach proposed by Benjamini and Hochberg was applied to t-tests; the threshold was adjusted for multiple comparisons [68].

Multivariate test statistics are reported using Wilks' lambda (Λ). In this context, smaller values of Λ indicate stronger effects of the predictor variable on the dependent variable. Partial eta squared ($\eta p^2$) is reported as effect size. Importantly, effect sizes were interpreted following conventional benchmarks defined by Cohen (1988): small effect ($\eta p^2 \approx .01$), medium effect ($\eta p^2 \approx .06$) and large effect ($\eta p^2 \geq .14$). Given the large sample size, statistical significance was interpreted alongside effect sizes to avoid overemphasising trivially small effects.

## Results

For each of the four tempi and two modes, mean ratings were first computed across the five songs. Inspection of composition-level descriptives revealed highly consistent patterns across all five stimuli for each emotion and tempo (Supporting Tables, S1 File).

These condition averages were then organised into the three emotional dimensions—sublimity, unease, and vitality. Results are reported separately for each dimension.

Table 1 provides a synoptic overview of the main effects and interactions of tempo and mode across the three emotional dimensions. Across analyses, the largest effects were observed for mode on unease and tempo on vitality. The effects of sublimity were consistently small. Given the large sample size, statistical significance is interpreted alongside effect sizes.

## Sublimity

A mixed-design ANOVA revealed a significant main effect of tempi on sublimity ratings, $F(3,1035)=6.00$, $p<.001$, $\eta p^2=.01$. Slower tempi were associated with higher sublimity ratings. However, the effect size was small.

There was also a significant main effect of mode, $F(1,1037)=18.32$, $p<.001$, $\eta p^2=.01$, with major mode yielding slightly higher sublimity ratings than minor mode. Again, the effect size was small.

The tempo × mode interaction reached significance, $F(3,1035)=4.40$, $p=.004$, $\eta p^2=.01$, although the magnitude of this interaction was small. The pattern indicated that tempo-related decreases in sublimity were comparable across modes.

**Table 1. Summary of main effects and interactions for tempo and mode across emotional dimensions.**

| Effect | Sublimity | Unease | Vitality |
|---|---|---|---|
| Tempo | F(3,1035)=6.00***, ηp²=.01 | F(3,1034)=9.90***, ηp²=.02 | F(3,1035)=35.10***, ηp²=.09 |
| Mode | F(1,1037)=18.32***, ηp²=.01 | F(1,1036)=189.51***, ηp²=.15 | F(1,1037)=135.86***, ηp²=.11 |
| Tempo×Mode | F(3,1035)=4.40**, ηp²=.01 | F(3,1035)=4.40**, ηp²=.01 | F(3,1035)=19.10***, ηp²=.05 |
| Tempo×Age | ns | F(3,1037)=6.50***, ηp²=.01 | ns |
| Mode×Age | F(1,1037)=18.18***, ηp²=.01 | F(1,1036)=179.57***, ηp²=.14 | ns |
| Other interactions† | small effects | small effects | small effects |

**\*\*\*** p<.001.

**\*\*** p<.01.

†Includes interactions involving gender, musicianship, and education level; all had small effect sizes (ηp²<.02).

ηp²=partial eta squared.

ns=non-significant.

A significant mode×age interaction was observed, F(1,1037)=18.18, p<.001, ηp²=.01, suggesting modest age-related differences in how mode influenced sublimity. No medium or large effects were observed for sublimity.

Overall, while tempo and mode significantly influenced sublimity ratings, all observed effects were small in magnitude (Fig 1). See supporting information tables (S1 File) for details.

## Unease

For unease ratings, a robust main effect of mode was observed, F(1,1036)=189.51, p<.001, ηp²=.15, representing a large effect. Compositions in the minor mode were consistently rated higher on unease than those in the major mode.

A significant main effect of tempi was also found, F(3,1034)=9.90, p<.001, ηp²=.02, although this effect was small. Unease ratings varied modestly across tempi.

The tempi×mode interaction was significant, F(3,1035)=4.40, p=.004, ηp²=.01, indicating that the influence of tempo differed slightly between modes, though the magnitude of this interaction was small.

A substantial mode×age interaction emerged, F(1,1036)=179.57, p<.001, ηp²=.14, approaching a large effect. This interaction indicated that age moderated the relationship between mode and unease, particularly in minor mode conditions.

Additional interactions involving tempo and age reached statistical significance (e.g., tempo×age), but all were small in magnitude (ηp²≤.02).

Taken together, unease ratings were primarily driven by mode, with tempo and demographic interactions contributing smaller effects (Fig 2). See supporting information tables (S1 File) for more details.

## Vitality

For vitality ratings, a strong main effect of tempi was observed, F(3,1035)=35.10, p<.001, ηp²=.09, representing a medium-to-large effect. Faster tempi were consistently associated with higher vitality ratings.

A substantial main effect of mode was also found, F(1,1037)=135.86, p<.001, ηp²=.11, representing a large effect. Major mode elicited higher vitality ratings than minor mode.

The tempi×mode interaction was significant, F(3,1035)=19.10, p<.001, ηp²=.05, indicating a moderate interaction. The increase in vitality with faster tempi was more pronounced in major mode.

Interactions involving age, gender, musicianship, and education reached statistical significance in several cases; however, all were small in magnitude (ηp²<.02).

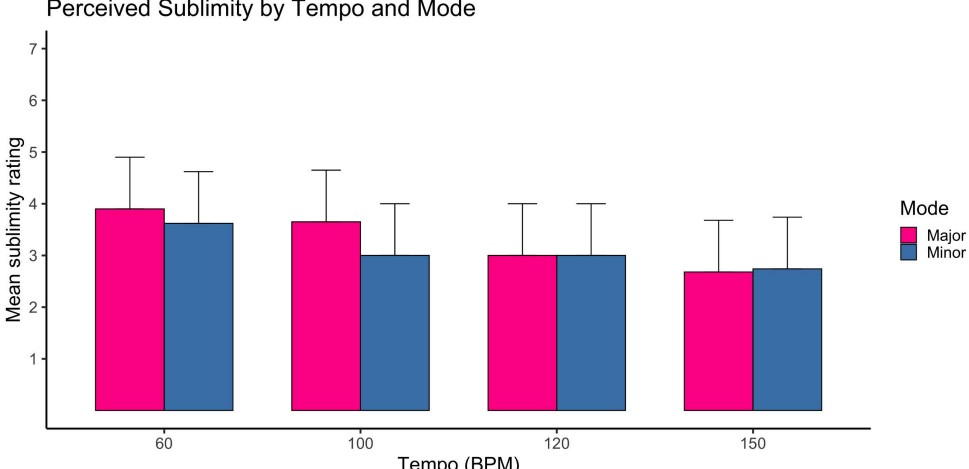

**Fig 1. Mean sublimity ratings as a function of tempi and mode.** Mean sublimity ratings are shown for four tempi conditions (60, 100, 120, 150 Beats Per Minute) in major and minor modes. Ratings were averaged across five structurally matched compositions within each tempi–mode condition. Error bars represent ±1 standard error of the mean. Higher values indicate stronger perceived sublimity. N = 1289 participants (complete cases vary slightly across conditions).

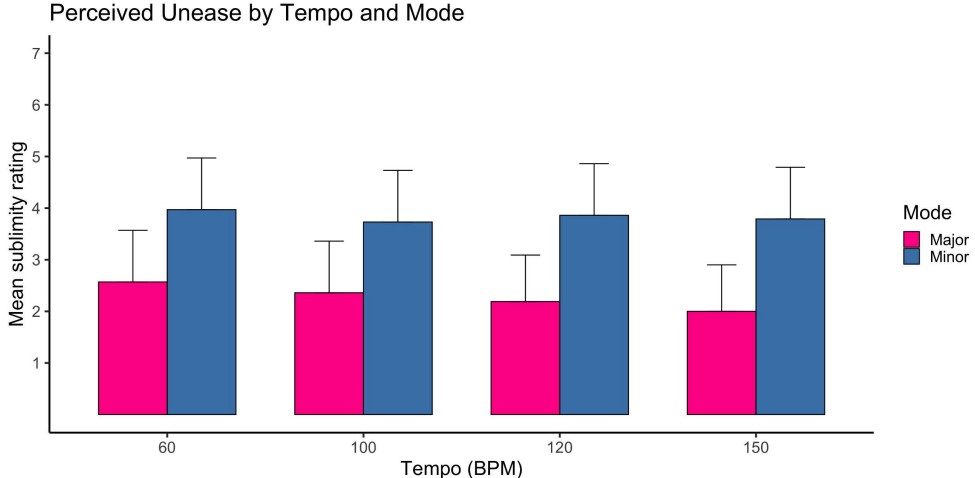

**Fig 2. Mean unease ratings as a function of tempi and mode.** Mean unease ratings are displayed for four tempi levels (60, 100, 120, 150 Beats Per Minute) in major and minor mode. Ratings were averaged across five compositions per tempi–mode condition. Error bars represent ±1 standard error of the mean. Higher values indicate greater perceived unease. N = 1289 participants. There was a pronounced main effect of mode.

Overall, vitality ratings were strongly influenced by tempo and mode, with tempo showing the clearest graded relationship (Fig 3). See supporting information tables (S1 File) for means and standard deviations.

**Age.** Given significant interactions involving age in the unease dimension, additional exploratory analyses were conducted. Participants were grouped into three age categories: 18–35 years, 36–59 years and 60 + years.

Separate one-way ANOVAs were conducted for selected tempo–mode combinations showing significant age interactions. Because 24 comparisons were examined, a Bonferroni-adjusted alpha level (α = .002) was applied.

Effect sizes (η²) are reported to contextualise statistically significant findings.

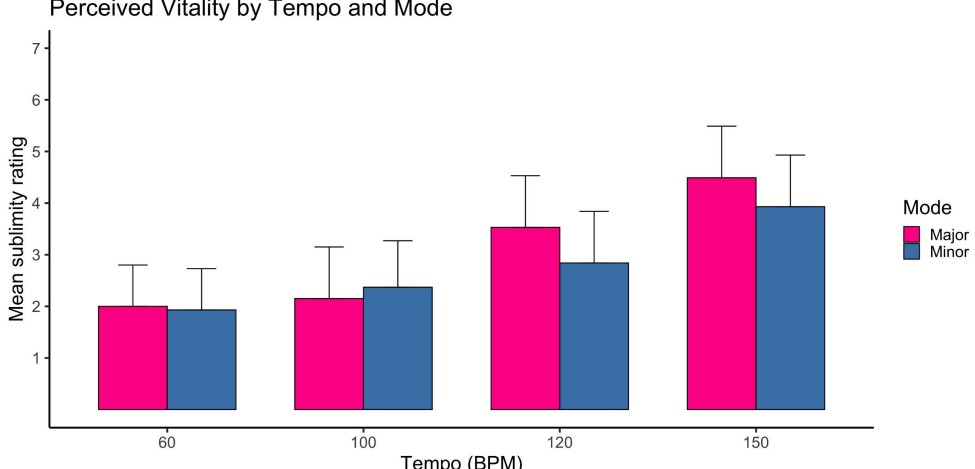

**Fig 3. Mean vitality ratings as a function of tempi and mode.** Mean vitality ratings are shown across four tempi conditions (60, 100, 120, 150 Beats Per Minute) in major and minor mode. Ratings were aggregated across five compositions within each condition. Error bars represent ±1 standard error of the mean. Higher values indicate greater perceived vitality. N = 1289 participants.

Given the pronounced mode × age interaction for unease, exploratory analyses were conducted across age groups. Significant differences were observed particularly for minor mode at 120 and 150 BPM, where younger participants reported higher unease ratings than older participants. These effects ranged from small to medium magnitude (η² up to .09).

For sublimity and vitality, age-related differences were generally small. See details of analysis in the supporting information tables (S1 File).

Visual representation of unease, 120 minor (Fig 4) and 150 minor (Fig 5), which were the only ones with a medium effect size, as defined by Cohen [69].

**Control stimuli.** To verify that emotional effects observed in the music conditions were not attributable merely to participation in an auditory task, paired-sample t-tests compared aggregated music ratings with aggregated control stimulus ratings. Control stimuli consisted of:

Three noise types (variants of white, pink, brown) and six neutral voice recordings.

For each emotional dimension, control ratings were averaged across all control stimuli. Comparisons were conducted separately for major and minor music conditions.

Bonferroni-adjusted significance thresholds were applied. Effect sizes (η²) were computed for all comparisons.

Paired-sample comparisons between aggregated music stimuli and control stimuli confirmed that emotional ratings differed systematically between music and non-musical sounds.

Music stimuli in both major and minor modes were rated significantly higher on sublimity and vitality than control stimuli (all p < .001). For unease, minor-mode music elicited higher ratings than controls, whereas major-mode music stimuli elicited lower unease ratings than controls.

Effect sizes for music versus control comparisons ranged from moderate to large. [68]. See the supporting information tables (S1 File) for a detailed analysis of all the second-order factors compared to control stimuli. A visual representation is presented (Fig 6).

## Discussion

Overall, the findings were consistent with our four hypotheses. Faster tempi were associated with significantly higher vitality ratings, with a medium-to-large effect size. Slower tempi were associated with higher sublimity ratings, although these

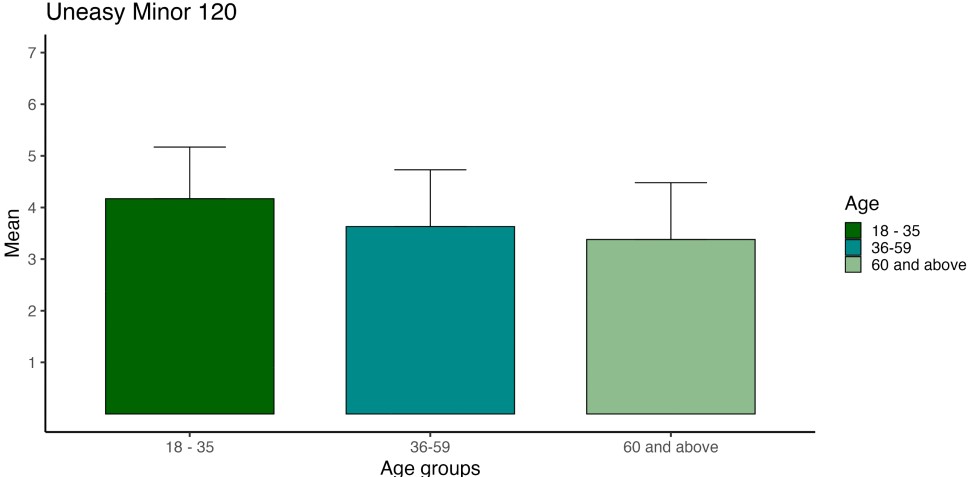

**Fig 4. Unease ratings by age group across tempi in 120 minor mode.** Mean unease ratings for minor-mode stimuli are displayed separately for three age groups (18–35, 36–59, 60+years) across four tempi levels. Ratings were averaged across five compositions per condition. Error bars represent ±1 standard error of the mean.

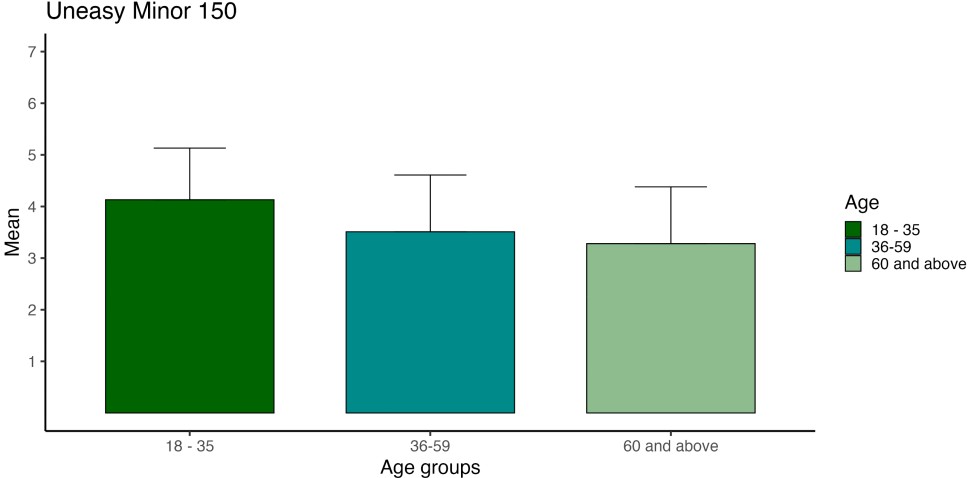

**Fig 5. Unease ratings by age group across tempi in 150 minor mode.** Mean unease ratings for minor-mode stimuli are displayed separately for three age groups (18–35, 36–59, 60+years) across four tempi levels. Ratings were averaged across five compositions per condition. Error bars represent ±1 standard error of the mean.

effects were small in magnitude. Minor mode stimuli elicited substantially higher unease ratings than major mode stimuli, representing the largest observed effect. These patterns were robust across the aggregated stimulus conditions, although the magnitude of effects varied across emotional dimensions. This finding is consistent with earlier research, which suggests that listeners often report experiencing negative emotions in music, such as sadness and tension, which is tied to the aesthetic appreciation rather than utilitarian goals [70].

Additionally, in our prior study, available as a preprint [62], which included tempi, mode, and dynamics as variables, tempi and mode similarly emerged as the primary influences, albeit with a smaller sample size. With the high sample size of the current study, our results had a 2.68% chance of incorrectly rejecting the null hypothesis. In the past century,

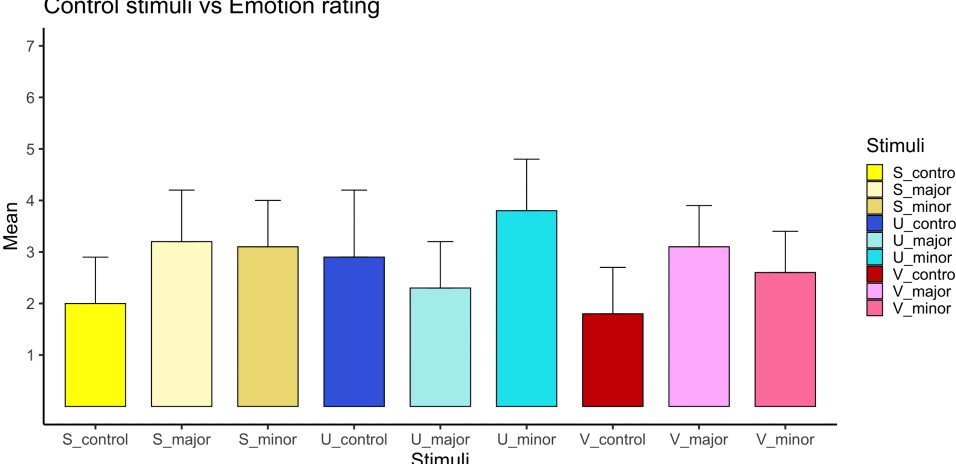

**Fig 6. Comparison of music and control stimuli across emotional dimensions.** Mean ratings for musical stimuli (aggregated across tempi and mode) and non-musical control stimuli (voice and noise) are shown for sublimity, unease, and vitality. Musical ratings were averaged across all tempi–mode conditions; control ratings were averaged across all control stimuli. Error bars represent ±1 standard error of the mean. N = 1289 participants.

psychological research tended to have small effect sizes, which often comes from small sample sizes [71]. Low power reduces the likelihood of finding a true effect [72]. With our study, we confirm previous research on emotion and tempi and mode, with high power, large sample size, and rigorously tested compositions and control stimuli.

In addition, we pilot tested our compositions, which has been suggested is important to control individual elements for emotional valence in music [34]. Yet only 1% of music stimuli is composed for studies, and only 3% of music stimuli is pilot tested [4]. We hope that our study contributes to the field by building on existing knowledge.

We implemented controls to address the possibility of compositions having a disproportionate effect, receiving higher ratings than control stimuli, except for major unease, where control stimuli received higher ratings. These findings might suggest that our music stimuli in a major mode were not rated high for unease, which aligns with our hypothesis and our other findings. Although the control stimuli may have also influenced the results to some extent, the use of rigorous control stimuli appears to be a beneficial methodological procedure.

Fast tempi have consistently been linked with joyful emotions [42], evoking a sensation of vitality, joyful activation, and power. This is in line with Gagnon & Peretz´s study, where it was reported that tempi and mode significantly contribute to happy/sad judgments, with tempi being more salient [73]. Similarly, fast tempi are associated with happiness, while slow tempi are linked to peacefulness [29]. Furthermore, patients with cochlear implants found that tempi were the most important parameter for rating music as happy when listening only with the Cochlear implant ear [35]. These studies, taken together with our findings, show that fast tempi are an important parameter in the perception of happiness/joy in music.

Comparing our study to others that do not employ the second-order factors of the GEMS-9 might pose challenges due to variations in emotional wordings. In the GEMS-9, emotional words associated with sublimity include wonder, transcendence, tenderness, nostalgia, and peacefulness. Previous research has demonstrated that slow tempi can evoke feelings of peace and tenderness [32]. This is in alignment with our study, where slow tempi were associated with sublimity ratings. Furthermore, fast tempi align with happiness and activation [32], which are subcategories of vitality. In our study, we found that fast tempi evoked ratings of vitality. The consistency across our study and previous research underscores the robust impact of tempi and mode on the perception of emotional content in music.

Furthermore, the CODA model encourages listeners to construct emotional experiences in response to music elements, such as tempi and mode [17]. There is a continuous appraisal process, which, in our study, is reflected in the changes of the ratings along with the changing tempi.

In our study, we found several significant differences related to gender, age, and education, although most of these had small effect sizes. It is crucial to acknowledge that small effect sizes indicate a relatively small impact on the results. With a sufficiently large sample, statistical methods will almost always demonstrate a significant difference unless there is no effect whatsoever [74].

For the current study, we aimed to measure perceived emotion, using a scale that was designed to measure both perceived and induced emotion. The music stimuli and the study were designed as part of a larger project, where both induction and perception would be measured. Participants were simply asked to give their opinions. We acknowledge that the current study might not be a pure perception study, and it was not intended to be so.

### Age

Most age differences in our study had small effects, except for unease in minor 120 and 150 BPM, which exhibited medium effects. The youngest age group (aged 18–35) had the highest ratings for compositions of unease in minor at 120 and 150 BPM, and the oldest age group (+60) had the lowest ratings for compositions of unease in minor at 120 and 150 BPM. These observations somewhat deviated from one study [26] where younger individuals showed more extreme reactions to music. Notably, our younger group demonstrated stronger ratings for unease minors at high tempi, suggesting potential age-related variations in the emotional ratings of unease. However, music likely operates at multiple levels, biological, psychological, and cultural [75], which means we may get emotional cues from tempi and mode, although there will always be individual differences, for example, between age groups.

Contrary to Peace and Halpern´s findings, our study somewhat aligns with another study [52], showing that adolescents and older adults tend to perceive music more positively compared to young and middle-aged adults. Additionally, a decrease in intense and contemporary music genre preferences with age and an increase in unpretentious and sophisticated music (genres from the MUSIC model [76]) has been shown [53]. Furthermore, older adults did not discriminate arousal differences for peaceful and threatening music compared to younger adults [77]. Similarly, compared to younger listeners, older participants rate consonant chords (typical for major mode) less pleasant and dissonant chords (typical for minor mode) more pleasant [78]. Middle-aged adults (mean age 47) rated sadness and fear lower than young adults (mean age 24) [53]. These results are somewhat like our findings, as unease represents sadness and fear.

Interestingly, other studies have found no effect on age and music perception [79,80]. These nuanced age-related differences underscore the complex interplay between age and emotional responses to music. It's also worth noting that our age groups did not have an equal number of participants, with the oldest age group only consisting of 180 participants, compared to the youngest group with over 600 participants, and the middle age group with around 400. As most of our age-related differences had small effect sizes, it might indicate an age sensitivity for tempi and mode, but more research is needed to explore this further.

### Limitations

One limitation of our study is the wording of the translated GEMS-9 scale in Norwegian. Using non-everyday words to describe emotions in Norwegian may have introduced a potential source of bias or misunderstanding. Additionally, a recent study [81] found correlations between rating music with emotional words and taste, which means we may not be constricted by language when attributing qualities to music. A valuable avenue for future research could involve introducing a dedicated potential source of bias or misunderstanding. Future research could involve a dedicated study evaluating and refining the emotional wording in the translated Norwegian version of GEMS-9, comparing it to the original English version of GEMS-9. Similar efforts have been undertaken for the German and English translations [82].

Additionally, the online nature of our study introduced limitations related to the control over participants´ environmental conditions. While participants were instructed to use headphones and take the survey in a quiet environment, the online setting

made it impossible to check if the instructions were followed. Factors such as changes in volume, headphone usage, and ambient noise could have influenced participants' experiences. Although efforts were made to ensure participants completed the survey in one sitting, we could not exclude them or the possibility that some participants left the page open on their computers for extended periods. These aspects highlight the inherent challenges and limitations associated with conducting online studies. However, another study [83] has pointed out the benefit of doing an online survey, such as higher ecological validity.

Another limitation is using the second-order factors of the GEMS-9. This is also a strength which makes our study more comparable to others that use dimensional models. However, in the confirmatory factor analysis correlation of the development of the GEMS-9, results indicated that the second-order factor unease was relatively low for the feeling of sadness, but perfect for tension [19] However, it has been shown that it is less common to feel negative emotions for music listening, such as sad, angry, tense, disgust, and guilt, compared to positive emotions such as happy, nostalgia, calm, loving, tender, and amused [84].

Using simple MIDI piano beats with constant rhythms, we reduced confounding music parameters variables, but they were less ecologically valid. Despite using a control design setting, this is also a limitation, and some participants expressed that they did not consider the stimulus music. While others expressed that they enjoyed the music very much.

Finally, the noises used for the control stimuli were only like brown, pink, and white noise, limited to a lower frequency range than the actual brown, pink, and white noise. This difference was discovered after the data had been collected and analysed. For the analysis, we combined all the noise stimuli and the voice recordings, which were control stimuli. The emotional ratings for control stimuli were significantly lower than for the music.

## Conclusion

In conclusion, our study delved into the intricate interplay of tempi and mode concerning the perception of emotion in original music stimuli. Our findings were aligned with previous research and our three research hypotheses. Fast tempi result in higher ratings for vitality, slower tempi yield higher sublimity ratings, and minor mode leads to higher ratings of unease. Though this is not a novel finding, our data were based on a large sample size, with high power, many large effect sizes, self-composed pilot-tested music, and various control stimuli, all fall under a high methodological standard for research.

It is also interesting that our research aligns with the CODA model to investigate tempi and mode with an emotional construct of appraisal and basic emotion ideas. Tempi and mode are significant factors in shaping how music stimuli is rated for sublimity, unease, and vitality. Minor mode and higher tempi also emerged as important for some age-related differences in uneasiness ratings.

For future research, it would be intriguing to conduct similar large-scale studies with different types of music stimuli, another music evaluation scale, and perhaps in different parts of the world to explore whether there exists a universal trend in the perception of tempi and mode in music. Future research should also strive to pilot test music and use control stimuli, and report the analysis of the control stimuli compared to the music.

## Supporting information

**S1 File. Supporting information, including access to musical and control stimuli, sheet music for all stimuli, and supplementary tables.**
(DOCX)

## Acknowledgments

A great thanks to Kjetil Vikene, who programmed the survey, and emphasised the importance of age differences. Thanks to Amalie Gloppen Norheim and Tina Emilie Johansen, who were excellent research assistants and helped with data collection. Thanks to the clinical psychology program students Kasper Bjørø, Elin Maria Kavlie Bryde, Kaja Kåstad Hauge,

Bernhard Vestby Edvardsen, Olav Erland, Randi Vaage Nordihus, Hani Kadri Vikebø, Sofie Follestad Kverkild, and Torgeir Jensen, who assisted with the data collection. Thanks to Lydia Brunvoll Sandøy and Tobias Bashevkin for letting me steal their voices.

## Author contributions

**Conceptualization:** Ulvhild Helena Færøvik.

**Data curation:** Ulvhild Helena Færøvik.

**Formal analysis:** Ulvhild Helena Færøvik.

**Funding acquisition:** Karsten Specht.

**Investigation:** Ulvhild Helena Færøvik.

**Methodology:** Ulvhild Helena Færøvik.

**Project administration:** Ulvhild Helena Færøvik.

**Resources:** Ulvhild Helena Færøvik.

**Supervision:** Karsten Specht.

**Validation:** Ulvhild Helena Færøvik.

**Visualization:** Ulvhild Helena Færøvik.

**Writing – original draft:** Ulvhild Helena Færøvik.

**Writing – review & editing:** Ulvhild Helena Færøvik, Karsten Specht.

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
