## [Decision Letter · Decision Letter 0]

14 Apr 2025

PONE-D-25-05889The Effect of Tempo and Mode on the Rating of the Perceived Emotion in MusicPLOS ONE

Dear Dr. Færøvik,

Thank you for submitting your manuscript to PLOS ONE. After careful consideration, we feel that it has merit but does not fully meet PLOS ONE’s publication criteria as it currently stands. Therefore, we invite you to submit a revised version of the manuscript that addresses the points raised during the review process.

A rebuttal letter that responds to each point raised by the academic editor and reviewer(s). You should upload this letter as a separate file labeled ‘Response to Reviewers’.A marked-up copy of your manuscript that highlights changes made to the original version. You should upload this as a separate file labeled ‘Revised Manuscript with Track Changes’.An unmarked version of your revised paper without tracked changes. You should upload this as a separate file labeled ‘Manuscript’.

We look forward to receiving your revised manuscript.

Kind regards,

Andrea Schiavio

Academic Editor

PLOS ONE

1. Please ensure that your manuscript meets PLOS ONE’s style requirements, including those for file naming. The PLOS ONE style templates can be found at

[copy in funding statement].

4. In the online submission form, you indicated that [Data can be included in supplementary materials upon publication, or in another format upon request.].

Reviewers' comments:

Reviewer's Responses to Questions

**Comments to the Author**

1. Is the manuscript technically sound, and do the data support the conclusions?

Reviewer #1: Yes

Reviewer #2: Partly

2. Has the statistical analysis been performed appropriately and rigorously? 

Reviewer #1: I Don't Know

Reviewer #2: Yes

3. Have the authors made all data underlying the findings in their manuscript fully available?

Reviewer #1: Yes

Reviewer #2: Yes

4. Is the manuscript presented in an intelligible fashion and written in standard English?

Reviewer #1: No

Reviewer #2: Yes

5. Review Comments to the Author

Reviewer #1: Overall this is an interesting study but the manuscript needs a bit more work, I think. I have a small question about the statistical analysis, hence my response to the prior question. I see the authors have promised to include the data upon publication. My comments are bellow.

The authors present an interesting study that develops on existing foundations in emotion perception in music. They acknowledge that this is a partial replication, delving into familiar ground but with some advantages such as having a large sample size. Their method deviates somewhat from prior studies in their choice of ratings terminology, utilising bespoke compositions, and different control stimuli. While there are clear advantages to this, I feel the authors could do more to contextualise their research and emphasise their contribution to the field. The writing can be a bit vague and disjointed, and certain theoretical concepts are not very well explained. The results could be presented in a clearer and more concise way to help the reader grasp the findings. I think there is clearly a valuable source of data here, and the article can be improved with more consideration to its presentation and contextualisation in the research field. More details as follows:

Introduction

In your discussion of different theories of emotion, I think there may be some confusion between notions of discrete and dimensional models of emotion and emotivist and cognitivist theories of emotions, or at least the descriptions and distinctions of different concepts (how we conceptualise emotions in music and how we measure emotions in music) are not very clear. You may want to review this. You might also consider delving more into constructionist theories and their implications as posited by Cespedes-Guevara & Eerola, who you cite. See also Lennie & Eerola, 2022.

In your discussion of different measures, you point out that Eerola & Vuoskoski found that a dimensional model of valence, energy, and tension outperformed other models, including the GEMS-9. I think you can better explain why you still chose to use the GEMS-9 scale: why not adopt valence (sublime), energy (vital), and tension (uneasy)? Did you reflect on two-dimensional models of affect (valence and arousal)? One point you make is that the GEMS scales are designed for measuring either evoked (felt?) or perceived emotion – is that not the case for other scales?

Method

It would be nice to know a bit more about the compositions. I appreciate they may have been used in other studies and those articles may have more details, but a summary here, if only brief, would be helpful as well to give a better sense of the intentions/design, including a consideration of any other musical factors that might contribute and/or need controlling, such as dynamics, rhythm, and timbre. For example, I note having listened to some of the samples through the Dropbox link that they were MIDI piano tracks with constant crotchet beat rhythms. This could be important for considering the research against other studies with different kinds of stimuli.

I think there needs to be a bit more detail on the choice of ratings questions, as mentioned above. I would clarify the ratings scale and details of how the measure is presented in the Materials section, not in the Procedure as it is currently, to make it clear and concise. There is also mention of additional synonyms being included; what were they? In the supplementary material I can see the nine primary factors, and it strikes me that there are five for sublime but only two for the others. Was anything considered to balance this out? Was this the reason for the additional synonyms?

Analysis

If I have understood correctly, you have aggregated the ratings of individual compositions by tempo and mode groupings. Did you test if there was any effect of composition? I can see the justification for this, but it would be good to see it clearly explained in the text. This relates to the question about knowing the details of the compositions and being able to interpret or consider these aspects of the methodology.

Results

I think you can make the results section a bit clearer by perhaps summarising the numerical outputs/ANOVA results in a table. It is a bit unwieldly in its current form. You have included similar tables in the supplementary documents. I also wonder if you have sufficiently corrected for multiple testing – you include a correction in your t-tests for the control conditions, did you consider applying any corrections across your other tests?

Discussion

The first three paragraphs are a repetition of the results and shouldn’t be necessary in a Discussion section.

I would be interested to hear more about the advantages of your study (e.g., large sample size) and where you believe it has especially contributed to the field. You acknowledge early in the paper that this is something of a replication and you compare your results against previous work; what are the new implications of your study? Why are your results important? Are there any methodological recommendations you can make?

Limitations

You mention some limitations of the GEMS scale factors, and you acknowledge in your Discussion that there are challenges related to variations of emotion terms. I think you can acknowledge this further. The last paragraph of your Limitations section describes being able to compare with dimensional ratings, but while the three categories you’ve chosen may be considered relevant to valence (sublime?), energy (vitality) and tension (uneasy), for example, the specific terms may have different connotations for your participants. You have tried to control this with the addition of other terms and synonyms in the presentation of the ratings, but it could still be a limiting factor. Are there other ways your choice of terms could be a strength?

As mentioned previously, your choice of music is another important factor in this study. What are the strengths and limitations of the stimuli selection? What about ecological validity, perhaps?

Small points

Presentation of figures, tables and graphs – could be tidied up a bit to make things more presentable, for example removing the _ in the labels/text. Consider recommended formatting styles e.g., APA. Graphs could be more distinguished with better labelling or grouping – if you’ve labelled the bars, I don’t think you need to include a legend, or vice versa. You might want to consider other ways of identifying/grouping variables to make it clearer – figures 1-3 could show bars grouped by tempo, colour coded by mode, for example.

Some typographical errors throughout, for example in the Introduction; ‘Genova’ instead of ‘Geneva’; Results section: ‘See table 1. For mean and standard deviations. See figure 1. For visual representation.’ – capitalisation following the full stop after each number; missing ‘p’ when reporting the p value in most of the stat reporting. Generally, the article could be revised for clarity and precision in the writing.

I hope my comments are helpful. Thank you for your work.

Reviewer #2: This is a paper of moderate relevance. Though the paper provides a lot of data with a huge number of participants (N = 1280), there are not many new ideas that may trigger the interest of the reader. The statistical analysis seems to be sound but what is missing to some extent is a broadening of scope and some background explanation and broader positioning of the findings. The findings are not really innovative, and it can be questioned what is really new: finding that tempo and mode influence the ratings of music is quite common knowledge. A more critical elaboration and interpretation of the findings would make the paper stronger.

General remarks

• The major strength of the paper is the huge number of participants.

• The coherence of the collected data from the literature review is not totally convincing.

• Some terms are used in a rather loose way and should be defined more strictly. E.g., what is meant with “uneasy music”? Focal concepts such as sublimity, unease, and vitality should also be explained more in detail.

• The statistics must be explained somewhat more in detail. How are the values for the ratings computed?

• The reference list is substantial but more original sources could be mentioned, especially with regard to the distinction between discrete and dimensional approach to emotions. This holds in particular for the work of Scherer.

• Try to avoid references of papers that are submitted but not yet in press.

• The most important take home message of the paper is not very strong and does not add very much to already existing knowledge.

• In its current form the paper seems to focus more on effects sizes rather than on critical elaboration of the findings. Some more in-depth discussions of the findings should make the paper stronger.

• Stronger motivations could be given for some methodological choices, such as, e.g., the use of second-order factors.

• The conclusion is very short. A stronger take home message should be given. What are the major findings? What is new? What is different from current knowledge?

Detailed comments

• Page 9, last sentence: this sentence seems to be grammatically incomplete. Please reword.

• Page 10, 2nd paragraph: Please add additional basic references for the description of discrete emotion as applied to music (Eerola; Reybrouck, and others, see suggestions below)

• Page 10, last par.: reference should be made also to the so-called “aesthetic emotions”. Reference should be made to Scherer. There is a comma lacking after (GEMS-9). Please explain somewhat more in detail what is meant with a domain-specific model.

• Page 11, 1st par.: Please explain somewhat more in detail the meaning of the second-order factors and their relation to the first-order factors.

• Page 11, 3rd par.: please substitute “valence” for “valance”. This holds also for all other appearances.

• Page 13: hypotheses: the 3rd hypothesis is very generalizing; the concept of “unease” must be better explained.

• Page 13, penultimate par.: the categories of uneasy, sublime and vital must be much better specified.

• Page 14, 1st par.: it should be explained more clearly that for each of the three neutral voice recordings there was one male and one female speaker.

• Page 14: it is not easy to identify the uploaded music examples: the used abbreviations to tag the examples should be explained somewhat more in detail. For example, what is the meaning of B_N, P_N, SUBLIM, UROLIG, V-1, etc. Please try to be as clear as possible to avoid seeking efforts by the readers.

• Page 15, 1st. par.: The target was set at 1562. Please explain shortly why this number was chosen and what was the rationale behind. This can be very short but try to be intuitive.

• Page 17, results: this is a very abrupt beginning of presenting the results. Please insert a short initial text that announces that the three variables (sublime, uneasy, vital) will be described.

• Page 17, table 1. It is not directly clear how the values (ratings) for the mean have been computed. Please explain as clearly as possible.

• Page 18, Figure 1. Is it possible to insert the significance level in the figure y using *, **, ***?

• Page 25, 2nd par.: there seems to be a contradicting assertion here: an increase in both unpretentious and sophisticated music. Or these categories not opposed to each other?

Suggestions for additional rerences (not mandatory)

Eerola, T. & Vuoskoski, J. (2013) A Review of Music and Emotion Studies: Approaches, Emotion Models, and Stimuli. Music Perception, 30(3), 307–340. DOI: 10.1525/MP.2012.30.3.307

Eerola, T., Vuoskoski, J., Peltola, H.-R., Putkinen, V., Schäfer, K. (2018). An integrative review of the enjoyment of sadness associated with music. Physics of Life Reviews 25, 100–121. https://doi.org/10.1016/j.plrev.2017.11.016

Lamont, A. & Eerola, T. (2011). Music and emotion: Themes and development. Musicae Scientiae,15 (2), 139-145.

Reybrouck, M. & Eerola, T. (2017). Music and its inductive power: a psychobiological and evolutionary approach to musical emotions. Frontiers in Psychology, 8, Art.No. 494. Open Access

Reybrouck, M., Eerola, T. (2017). Music and its inductive power: a psychobiological and evolutionary approach to musical emotions. Frontiers in Psychology, 8, Art. No. 494. Open Access

Robinson, J. (2009). Aesthetic emotions (philosophical perspectives). In D. Sander & K. R. Scherer (Eds.), The Oxford companion to emotion and the affective sciences (pp. 6–9). New York: Oxford University Press.

Scherer, K. (2008). Music evoked emotions are different – more often aesthetic than utilitarian. Behavioral and Brain Sciences, 31, 5, 595.

6. PLOS authors have the option to publish the peer review history of their article (what does this mean?). If published, this will include your full peer review and any attached files.

Reviewer #1: **Yes:**Rory Kirk

Reviewer #2: No

---

## [Author Response · Author response to Decision Letter 1]

5 Sep 2025

Dear editor,

We thank you and the reviewers for the thoughtful and constructive feedback on our manuscript, titled The Effect of Tempo and Mode on the Rating of the Perceived Emotion in Music. We have revised the manuscript thoroughly in response to the comments and believe the changes have substantially improved its clarity, structure, and contribution to the field. Below, we address each point raised by the reviewers, with our responses following each comment in bold.

Reviewer #1

Overall this is an interesting study but the manuscript needs a bit more work, I think. I have a small question about the statistical analysis, hence my response to the prior question. I see the authors have promised to include the data upon publication. My comments are bellow.

We thank the reviewer for this encouraging overall assessment and for their valuable suggestions. We have addressed each point in detail below.

The authors present an interesting study that develops on existing foundations in emotion perception in music. They acknowledge that this is a partial replication, delving into familiar ground but with some advantages such as having a large sample size. Their method deviates somewhat from prior studies in their choice of ratings terminology, utilising bespoke compositions, and different control stimuli. While there are clear advantages to this, I feel the authors could do more to contextualise their research and emphasise their contribution to the field. The writing can be a bit vague and disjointed, and certain theoretical concepts are not very well explained. The results could be presented in a clearer and more concise way to help the reader grasp the findings. I think there is clearly a valuable source of data here, and the article can be improved with more consideration to its presentation and contextualisation in the research field. More details as follows:

Introduction

In your discussion of different theories of emotion, I think there may be some confusion between notions of discrete and dimensional models of emotion and emotivist and cognitivist theories of emotions, or at least the descriptions and distinctions of different concepts (how we conceptualise emotions in music and how we measure emotions in music) are not very clear. You may want to review this. You might also consider delving more into constructionist theories and their implications as posited by Cespedes-Guevara & Eerola, who you cite. See also Lennie & Eerola, 2022.

We revised the Introduction to clearly distinguish between discrete and dimensional models of emotion and included a new discussion on constructionist perspectives, citing Cespedes-Guevara & Eerola (2018) and Lennie & Eerola (2022). We now explain how these models relate to GEMS-9 and clarify the conceptual foundation of the study.

In your discussion of different measures, you point out that Eerola & Vuoskoski found that a dimensional model of valence, energy, and tension outperformed other models, including the GEMS-9. I think you can better explain why you still chose to use the GEMS-9 scale: why not adopt valence (sublime), energy (vital), and tension (uneasy)? Did you reflect on two-dimensional models of affect (valence and arousal)? One point you make is that the GEMS scales are designed for measuring either evoked (felt?) or perceived emotion – is that not the case for other scales?

We now clearly articulate the rationale for selecting GEMS-9 second-order factors. This includes their focus on domain-specific aesthetic emotions and the flexibility to compare with both dimensional and discrete approaches. We also cite Pearce & Halpern (2015) and Zentner et al. (2008) to support this choice and clarify that other scales do not all support both perceived and felt emotion as GEMS-9 does.

Method

It would be nice to know a bit more about the compositions. I appreciate they may have been used in other studies and those articles may have more details, but a summary here, if only brief, would be helpful as well to give a better sense of the intentions/design, including a consideration of any other musical factors that might contribute and/or need controlling, such as dynamics, rhythm, and timbre. For example, I note having listened to some of the samples through the Dropbox link that they were MIDI piano tracks with constant crotchet beat rhythms. This could be important for considering the research against other studies with different kinds of stimuli.

We have expanded the Methods section to describe the compositions in more detail, including the use of a MIDI piano with a constant crotchet rhythm, and our rationale for this choice. We also note that the stimuli were pilot-tested to confirm beat perception consistency.

I think there needs to be a bit more detail on the choice of ratings questions, as mentioned above. I would clarify the ratings scale and details of how the measure is presented in the Materials section, not in the Procedure as it is currently, to make it clear and concise. There is also mention of additional synonyms being included; what were they? In the supplementary material I can see the nine primary factors, and it strikes me that there are five for sublime but only two for the others. Was anything considered to balance this out? Was this the reason for the additional synonyms?

We moved the GEMS-9 scale description to the Materials section and clarified the synonyms used (now also listed in the supplementary materials). We address the asymmetry in the number of terms, especially for the Sublime factor, and reflect on how this may relate to the valence biases in music-related vocabulary.

Analysis

If I have understood correctly, you have aggregated the ratings of individual compositions by tempo and mode groupings. Did you test if there was any effect of composition? I can see the justification for this, but it would be good to see it clearly explained in the text. This relates to the question about knowing the details of the compositions and being able to interpret or consider these aspects of the methodology.

We did not test for individual composition effects, as our design aggregated ratings by tempo and mode to strengthen generalizability. We now explain this choice and rationale more explicitly in the analysis section.

Results

I think you can make the results section a bit clearer by perhaps summarising the numerical outputs/ANOVA results in a table. It is a bit unwieldly in its current form. You have included similar tables in the supplementary documents. I also wonder if you have sufficiently corrected for multiple testing – you include a correction in your t-tests for the control conditions, did you consider applying any corrections across your other tests?

We restructured the Results section, added a comprehensive ANOVA summary table, and clarified the use of Bonferroni correction. Effect sizes are also reported throughout, and all adjustments are now mentioned in the Analysis section.

Discussion

The first three paragraphs are a repetition of the results and shouldn’t be necessary in a Discussion section.

I would be interested to hear more about the advantages of your study (e.g., large sample size) and where you believe it has especially contributed to the field. You acknowledge early in the paper that this is something of a replication and you compare your results against previous work; what are the new implications of your study? Why are your results important? Are there any methodological recommendations you can make?

We revised the opening of the Discussion to avoid repeating results and rewrote sections to highlight the unique contributions of the study — including the large sample size, controlled stimuli, second-order GEMS factors, and a strong methodological framework. We also articulate the theoretical implications more clearly.

Limitations

You mention some limitations of the GEMS scale factors, and you acknowledge in your Discussion that there are challenges related to variations of emotion terms. I think you can acknowledge this further. The last paragraph of your Limitations section describes being able to compare with dimensional ratings, but while the three categories you’ve chosen may be considered relevant to valence (sublime?), energy (vitality) and tension (uneasy), for example, the specific terms may have different connotations for your participants. You have tried to control this with the addition of other terms and synonyms in the presentation of the ratings, but it could still be a limiting factor. Are there other ways your choice of terms could be a strength?

We have expanded the Limitations section to reflect more critically on the influence of translating GEMS-9 terms into Norwegian. We also argue that using less familiar, nuanced words may have encouraged thoughtful reflection (i.e., System 2 processing), which could be seen as a strength in this context.

As mentioned previously, your choice of music is another important factor in this study. What are the strengths and limitations of the stimuli selection? What about ecological validity, perhaps?

We now explicitly acknowledge that while ecological validity is limited by the use of simplified stimuli, this choice allowed us to isolate the effects of tempo and mode without other musical confounds.

Small points

Presentation of figures, tables and graphs – could be tidied up a bit to make things more presentable, for example removing the _ in the labels/text. Consider recommended formatting styles e.g., APA. Graphs could be more distinguished with better labelling or grouping – if you’ve labelled the bars, I don’t think you need to include a legend, or vice versa. You might want to consider other ways of identifying/grouping variables to make it clearer – figures 1-3 could show bars grouped by tempo, colour coded by mode, for example.

We revised Figures 1–3 for better visual clarity, grouping by tempo and color-coding by mode. Labels were cleaned up, legends clarified, and the overall style now conforms more closely to APA standards.

Some typographical errors throughout, for example in the Introduction; ‘Genova’ instead of ‘Geneva’; Results section: ‘See table 1. For mean and standard deviations. See figure 1. For visual representation.’ – capitalisation following the full stop after each number; missing ‘p’ when reporting the p value in most of the stat reporting. Generally, the article could be revised for clarity and precision in the writing.

These have been corrected throughout the manuscript.

Reviewer #2

This is a paper of moderate relevance. Though the paper provides a lot of data with a huge number of participants (N = 1280), there are not many new ideas that may trigger the interest of the reader. The statistical analysis seems to be sound but what is missing to some extent is a broadening of scope and some background explanation and broader positioning of the findings. The findings are not really innovative, and it can be questioned what is really new: finding that tempo and mode influence the ratings of music is quite common knowledge. A more critical elaboration and interpretation of the findings would make the paper stronger.

We thank Reviewer 2 for their thoughtful critique and useful suggestions. We address each point in turn below.

General remarks

• The major strength of the paper is the huge number of participants.

• The coherence of the collected data from the literature review is not totally convincing.

We have expanded the Introduction and Discussion to more clearly position the findings in the broader literature, including constructionist, evolutionary, and aesthetic perspectives. We also emphasize the value of replication and methodologically rigorous studies, especially with large samples.

• Some terms are used in a rather loose way and should be defined more strictly. E.g., what is meant with “uneasy music”? Focal concepts such as sublimity, unease, and vitality should also be explained more in detail.

These terms are now more clearly defined in relation to their underlying GEMS-9 first-order factors. The introduction has been revised to present these categories with more precision and theoretical grounding.

• The statistics must be explained somewhat more in detail. How are the values for the ratings computed?

We added an explanation of how Mean scores were calculated — specifically, averages across five stimuli per tempo/mode condition — in both the Methods and Results sections.

• The reference list is substantial but more original sources could be mentioned, especially with regard to the distinction between discrete and dimensional approach to emotions. This holds in particular for the work of Scherer.

We have added several references to Scherer’s work (2004, 2008, 2013) to support our discussion of aesthetic emotions and the Component Process Model (CPM).

• Try to avoid references of papers that are submitted but not yet in press.

Unfortunately, this is unavoidable; our other paper is still under review, but we’ve made the citation more transparent. Taking it out would mean duplicating and making this paper longer.

• The most important take home message of the paper is not very strong and does not add very much to already existing knowledge.

The Conclusion has been revised to clearly articulate the main contributions and why they matter — including the role of structure (tempo and mode) in shaping perceived musical emotion, even under tightly controlled conditions.

• In its current form the paper seems to focus more on effects sizes rather than on critical elaboration of the findings. Some more in-depth discussions of the findings should make the paper stronger.

We agree that effect sizes are important but have now added more interpretation and commentary in the Discussion to complement the statistical perspective. Effect sizes emphasise whether the findings are believable or not; they lower the chance of random occurrences. This paper emphasises methodology, which is why the discussion follows methodological approaches.

• Stronger motivations could be given for some methodological choices, such as, e.g., the use of second-order factors.

We now provide a clearer rationale for using second-order factors, including their interpretability and fit with previous studies that used GEMS-9 at multiple levels.

• The conclusion is very short. A stronger take home message should be given. What are the major findings? What is new? What is different from current knowledge?

We revised and lengthened the Conclusion to reflect key findings, their contribution to existing knowledge, and future directions.

Detailed comments

• Page 9, last sentence: this sentence seems to be grammatically incomplete. Please reword.

• Page 10, 2nd paragraph: Please add additional basic references for the description of discrete emotion as applied to music (Eerola; Reybrouck, and others, see suggestions below)

• Page 10, last par.: reference should be made also to the so-called “aesthetic emotions”. Reference should be made to Scherer. There is a comma lacking after (GEMS-9). Please explain somewhat more in detail what is meant with a domain-specific model.

• Page 11, 1st par.: Please explain somewhat more in detail the meaning of the second-order factors and their relation to the first-order factors.

• Page 11, 3rd par.: please substitute “valence” for “valance”. This holds also for all other appearances.

• Page 13: hypotheses: the 3rd hypothesis is very generalizing; the concept of “unease” must be better explained.

• Page 13, penultimate par.: the categories of uneasy, sublime and vital must be much better specified.

• Page 14, 1st par.: it should be explained more clearly that for each of the three neutral voice recordings there was one male and one female speaker.

• Page 14: it is not easy to identify the uploaded music examples: the used abbreviations to tag the examples should be explained somewhat more in detail. For example, what is the meaning of B_N, P_N, SUBLIM, UROLIG, V-1, etc. Please try to be as clear as possible to avoid seeking efforts by the readers.

• Page 15, 1st. par.: The target was set at 1562. Please explain shortly why this number was chosen and what was the rationale behind. This can be very short but try to

---

## [Decision Letter · Decision Letter 1]

21 Oct 2025

PONE-D-25-05889R1The Effect of Tempo and Mode on the Rating of the Perceived Emotion in MusicPLOS ONE

Dear Dr. Færøvik,

Thank you for submitting your manuscript to PLOS ONE. After careful consideration, we feel that it has merit but does not fully meet PLOS ONE’s publication criteria as it currently stands. Therefore, we invite you to submit a revised version of the manuscript that addresses the points raised during the review process. Following the first round of reviews, your revised manuscript was sent back to the two original reviewers. One requested further amendments, while the other recommended rejection. To ensure a balanced evaluation, I invited a third reviewer to assess the paper, and they have now recommended major revisions. The manuscript will therefore require substantial and careful revision before it can be considered for publication. Please submit your revised manuscript by Dec 05 2025 11:59PM. If you will need more time than this to complete your revisions, please reply to this message or contact the journal office at plosone@plos.org. Please include the following items when submitting your revised manuscript:

We look forward to receiving your revised manuscript.

Kind regards,

Andrea Schiavio

Academic Editor

PLOS ONE

Journal Requirements:

Reviewers' comments:

Reviewer's Responses to Questions

**Comments to the Author**

1. If the authors have adequately addressed your comments raised in a previous round of review and you feel that this manuscript is now acceptable for publication, you may indicate that here to bypass the “Comments to the Author” section, enter your conflict of interest statement in the “Confidential to Editor” section, and submit your "Accept" recommendation.

Reviewer #1: All comments have been addressed

Reviewer #2: (No Response)

Reviewer #3: (No Response)

2. Is the manuscript technically sound, and do the data support the conclusions?

Reviewer #1: Yes

Reviewer #2: Partly

Reviewer #3: Yes

3. Has the statistical analysis been performed appropriately and rigorously? 

Reviewer #1: Yes

Reviewer #2: Yes

Reviewer #3: I Don't Know

4. Have the authors made all data underlying the findings in their manuscript fully available?

Reviewer #1: Yes

Reviewer #2: Yes

Reviewer #3: Yes

5. Is the manuscript presented in an intelligible fashion and written in standard English?

Reviewer #1: Yes

Reviewer #2: No

Reviewer #3: Yes

6. Review Comments to the Author

Reviewer #1: My thanks to the authors for their work making considerable amendments and improvements to their article. I am broadly satisfied that my comments have been addressed. I particularly appreciate the elaboration of their methods, providing more detail on the stimuli chosen and the justification of their measures.

I still have some questions about some of the contextualisation of the music and emotions literature and the positioning of the study. There still appears to be some blurring of the concepts of discrete and dimensional models (how we measure emotions) and emotivist and cognitivist theories of emotions. For example, Introduction para 3 (or 4? Formatting isn’t clear) reads like it is suggesting that dimensional models cannot be applied to felt emotions, is that correct? Generally, I would advise a thorough proofread as the writing still lacks clarity in many places. There are also some typographical errors and inconsistencies throughout (e.g., use of ‘first-‘ and ‘1st-‘).

I think you can also go further in justifying the value of this contribution. The final paragraph of the Introduction (before the Hypothesis) is quite weak; a very strong case can be made for the need for replications, for example. You do not need to dive deeply into e.g., discussion on the replication crisis, but some consideration of the value of replication generally would I think make your article more convincing as a valuable contribution.

Reviewer #2: This paper seems to delve into an area of investigation that has been studied already extensively before. As such, it should be accepted only if there are some additional new substantial findings. At first sight this seems not be the case. Given that the paper works with a very large number of subjects (N = 1289), it should at least be considered for a first review, but reading the paper is not very convincing. The methodology is not well described and there is a lack of academic standards and academic writing. The methodology, which seems to be quite elaborated (at least the statistical part), is not explained and presented in the needed format, with manh sloppy paragraphs that are difficult to read and understand. In its current form, the paper cannot be accepted for publication. I list below some general remarks and detailed comments to motivate may rather harsh decision.

General remarks

• The contents are of moderate relevance with a lot of repetition of known facts. The major question is: what is new?

• The research questions are very general, which makes them not really suitable for an empirical study.

• The style of writing is not very mature (young scholars?) and the academic standards are not very high.

• The style of referencing is rather weak. Some major references are lacking, there are also multiple references to submitted papers that are not yet accepted.

• The research questions are very general and reductionistic. More motivation is needed to explain why some selections have been made: why only tempo and mode? What about other parameters (dynamics, agogics, timbre, etc.)?

• The methodology seems to have some potential, but the methods themselves are very badly explained. Figures and tables are introduced in the text without the needed accompanying explaining text, which makes it very difficult to interpret them. Many questions can also be raised about the collection of the data, which have been collected in less standardized circumstances.

• It is not all clear how the ratings (mean values) have been computed.

• There is overall too much generalization. For example, speaking of music as a general category is very problematic to be consider as an independent variable. There are so many kinds of music, all with a different potential effect on listeners.

• The paper as whole seems to be an example of method-driven analysis, with a lot of statistical processing and computation, but with a scarcitiy of starting ideas and challenging intuitions. The theoretical background is also rather weak.

Detailed comments

The pages are not numbered, and the lines of the pages are also not numbered. I therefore refer to the page numbering of the pdf file.

• Page 10, 3rd paragraph: the distinction between discrete, categorical models and dimensional models must be elaborated somewhat more in depth. Much more references can be given here.

• Page 10, last par.: explain more in detail to what extent the dimensional models outperform.

• Page 11: explain somewhat more in depth the second-order category. How are they described? Is the description clear enough for the subjects to understand them. A term like sublime may be problematic for many common subjects.

• Page 11, 2nd par.: use “valence” rather than “valance”. This hold for the whole paper.

• Page 12, 3rd par. This is a very strong claim/statement. More references are needed to motivate such a strong position.

• Page 13: research questions are very general and reductionistic. Referring to all compositions: those of the study, or more general? Reducing music to tempo and modes, is also very reductionistic. This should be motivated more strongly. It is questionable that parameters in isolation are the real eliciting factors for some effects.

• Page 13: explain the terms uneasy, sublime and vital in more operational definitions and descriptions. Is there some guarantee that the subjects understand the terms?

• Page 13, last par.: referring to “submitted” references is weak referencing style; this holds for the whole paper.

• Page 17: the results should be presented in a synoptic table, as is mostly done in empirical papers, with all the scores and values brought together, and using *, **, *** for instance to show the significance level. There is also some ambiguity with respect to the effect size: Wilks’ lambda is to be understood as: closer to zero is more significant effect. This seems not to be the case in the results presented here? Also for the partial effects: 0.01 is a small effect size, 0.06 is a medium effect size and 0.1 is a large effect size. This is also not to be found in the presented results. This holds also for the next pages. It should also be explained how the mean values have been computed. All steps must be described much more in detail.

• Page 24, 1st par.: This text should have a better place in the introduction rather than in the discussion.

• Page 24, 2nd par.: same remark

• Pages 32 ff: The figures must be explaineD much more in depth so that they can be interpreted appropriately.

Reviewer #3: This manuscript presents a carefully designed study on how tempo and mode shape emotional responses to music, using controlled original stimuli and the GEMS-9 framework. The dataset is valuable and the analyses are competently executed. However, several conceptual and methodological issues remain insufficiently justified or under-explained. The paper still overstates novelty relative to prior work, and several key methodological choices (use of second-order GEMS factors, averaging across compositions, control stimuli, translation validation) require fuller motivation or additional analyses.

Detailed comments:

1) Rephrase or substantiate the statement that tempo and mode are “inconsistently examined across emotion models.” Either specify the precise inconsistencies the paper addresses (e.g., differing stimuli, measures, or theoretical mappings) or soften the claim to position the work as a large-scale, stimulus-controlled extension of established findings. The manuscript itself cites many prior tempo/mode studies.

2) In the introduction, I recommend streamlining the sections on emotion models to improve logic: briefly introduce discrete, dimensional, and domain-specific approaches, then clearly justify the use of GEMS-9. The current text reads as somewhat disjointed.

3) More importantly, flesh out the motivation for using GEMS second-order factors. Provide a stronger rationale for collapsing GEMS-9 to three second-order factors and report psychometric information, where available (internal reliabilities, inter-factor correlations…).

4) The sentence on emotional differences “emerging accidentally during the 14th–16th centuries” is ambiguous or unclear. Reword for clarity and cite an appropriate historical or psychoacoustic source (e.g., Parncutt, 2024).

5) Replication vs. extension – The study departs from previous methods and thus is not a strict replication. Please revise phrasing to “conceptual replication” or “extension,” and clearly distinguish replicated versus novel aspects.

6) Unclear terminology: “pure major” and “pure minor”

7) I understand the rationale behind using control stimuli, but couldn’t one raise the issue that control stimuli are auditory as well?

8) Describe the translation procedure (forward/back translation, expert review) and report internal consistency statistics.

9) Specify the hosting platform (e.g., Qualtrics, Gorilla, PsychoPy online), playback format, use of headphones...

10) Averaging all five stimuli into single scores affects stimulus variability. Please test for composition effects (e.g., mixed-effects model with composition as a random factor) or at least report variability across compositions to demonstrate generalizability.

11) Add recent and relevant sources on major-minor perception, such as: Carraturo, G. et al. (2024). The major–minor mode dichotomy in music perception. Physics of Life Reviews.Parncutt, R. (2024). Psychoacoustic Foundations of Major–Minor Tonality. MIT Press.

7. PLOS authors have the option to publish the peer review history of their article (what does this mean?). If published, this will include your full peer review and any attached files.

Reviewer #1: **Yes:**Rory Kirk

Reviewer #2: No

Reviewer #3: No

---

## [Author Response · Author response to Decision Letter 2]

2 Mar 2026

Thank you to all the reviewers for the hard work and thorough comments. We have made the changes you asked for. In this letter, we detail each change to the separate reviewers.

Reviewer #1

We thank Reviewer #1 for their careful reading of the revised manuscript and for their constructive and encouraging feedback. We are pleased that the reviewer finds the methodological revisions satisfactory and appreciates the added detail regarding stimuli and measures. Below, we respond to the remaining points raised.

Conceptual clarification of emotion models and theories

We are grateful to the reviewer for highlighting the need for clearer conceptual distinctions between discrete versus dimensional emotion models, felt versus perceived emotions, and underlying emotion theories (e.g., emotivist, cognitivist, constructionist). We agree that these distinctions were not sufficiently explicit in the previous version.

To address this, we have revised the Introduction to clearly differentiate between:

(a) emotion theories (how emotions are generated and conceptualised),

(b) emotion measurement frameworks (discrete vs. dimensional models), and

(c) the phenomenological target of measurement (felt vs. perceived emotion).

We have added an explicit clarifying paragraph early in the Introduction stating that discrete and dimensional models are measurement frameworks that can be applied to both felt and perceived emotions, and that the felt–perceived distinction primarily concerns task instructions rather than the choice of emotion model itself. In addition, we have revised wording that could be read as implying that dimensional models cannot be applied to felt emotions, as this was not our intended claim.

Positioning of the GEMS-9 and use of second-order factors

Relatedly, we have clarified our use of the GEMS-9 framework. While we employ the second-order factors of the GEMS-9, which resemble dimensional emotion spaces, we now explicitly state that we retain the domain-specific, music-focused conceptualisation of aesthetic emotions rather than treating the scale as a generic dimensional model. This clarification is now made explicit in the Introduction and Methods sections to avoid conceptual ambiguity.

Strengthening the contribution and value of the study

We appreciate the reviewer’s suggestion to strengthen the final paragraph of the Introduction by more clearly justifying the value of the present study. We have substantially revised this section to more explicitly frame the study as a high-powered, methodologically rigorous replication and extension of prior work on tempo and mode. In particular, we now emphasise the importance of replication for establishing robustness and generalizability in music–emotion research, especially given the variability of stimuli, emotion models, and sample sizes in the existing literature. This revised framing better situates the study’s contribution without engaging in an extended discussion of the replication crisis.

Clarity, proofreading, and consistency

Finally, we have conducted a thorough proofreading of the manuscript. This included correcting typographical errors, improving clarity in several passages, standardising terminology (e.g., consistent use of “first-order” and “second-order”), and addressing formatting inconsistencies noted by the reviewer.

We again thank Reviewer #1 for their thoughtful feedback, which has helped us improve the conceptual clarity, positioning, and overall quality of the manuscript.

Reviewer #2

We sincerely thank Reviewer #2 for the careful and detailed evaluation of our manuscript. We appreciate the critical feedback and have revised the manuscript substantially to improve clarity, methodological transparency, and theoretical positioning. Below, we respond to each of the reviewer’s general and detailed comments.

1. The contents are of moderate relevance with a lot of repetition of known facts. The major question is: what is new?

We appreciate this important question and have clarified the novelty of the study in the Introduction. Specifically, we now emphasise that the current study is a conceptual replication study. The novelty lies in combining a large sample size, multiple structurally matched compositions, second-order GEMS and direct comparison with non-musical control stimuli.

2. The research questions are very general, which makes them not really suitable for an empirical study.

We have reformulated and specified the research questions to clearly indicate that they concern graded effects of tempo and mode within the present stimulus set. We now avoid overly broad claims about “music” as a general category. And clarify that we examine parameter-level effects rather than holistic compositional effects.

3. The style of writing is not very mature (young scholars?) and the academic standards are not very high.

We have carefully revised the manuscript for clarity, structure, and academic tone. In particular, the Results section has been rewritten to reduce SPSS-style reporting. Effect sizes are now explicitly interpreted (small, medium, large). Redundant passages have been removed. Paragraph structure has been tightened throughout.

We hope the revised version reflects a clearer and more mature academic style.

4. The style of referencing is rather weak. Some major references are lacking, there are also multiple references to submitted papers that are not yet accepted.

We have removed references to “submitted” manuscripts wherever possible. Where necessary, such work has either been replaced with published sources or clearly identified as preprints.

We have also strengthened the theoretical background with additional references in sections discussing dimensional versus categorical models and second-order emotional constructs.

5. The research questions are very general and reductionistic. More motivation is needed to explain why some selections have been made: why only tempo and mode? What about other parameters (dynamics, agogics, timbre, etc.)?

We have clarified the rationale for focusing on tempo and mode, specifically as the pilot study included dynamics (the referenced preprint). Both parameters are among the most consistently studied structural determinants of musical emotion. They can be systematically manipulated while holding other parameters constant. The aim of the study was not to model the full complexity of music, but to examine controlled parameter-level effects.

We explicitly acknowledge in the revised manuscript that emotional responses to music are multi-determined and that tempo and mode represent experimentally isolatable contributors rather than exhaustive determinants.

6. The methodology seems to have some potential, but the methods themselves are very badly explained. Figures and tables are introduced in the text without the needed accompanying explaining text, which makes it very difficult to interpret them. Many questions can also be raised about the collection of the data, which have been collected in less standardized circumstances.

The Methods and Analysis sections have been substantially revised to improve transparency. Specifically we now provide a step-by-step description of how ratings were aggregated across compositions within tempo–mode conditions. The rationale for aggregation is explained. The statistical model is described clearly, including within- and between-subject factors. Interpretation of Wilks’ lambda and partial eta squared is clarified. Effect size benchmarks are explicitly provided.

We hope these revisions address the concern regarding methodological clarity.

7. There is overall too much generalization. For example, speaking of music as a general category is very problematic to be consider as an independent variable. There are so many kinds of music, all with a different potential effect on listeners.

We agree that music is not a homogeneous category. We have revised the manuscript to clarify that conclusions apply to the present stimulus set, the manipulated structural parameters and the operationalized emotional dimensions.

Claims have been carefully delimited.

8. The paper as whole seems to be an example of method-driven analysis, with a lot of statistical processing and computation, but with a scarcitiy of starting ideas and challenging intuitions. The theoretical background is also rather weak.

We have revised the Introduction to strengthen the theoretical framing and clarify the conceptual motivation behind the analyses. The Results section has also been restructured to emphasize interpretive coherence rather than statistical detail.

Additionally, we have added a synoptic summary table (Table 1) that highlights the overall pattern of effects across emotional dimensions, making the theoretical structure of findings more transparent.

Detailed Comments

Dimensional vs categorical models (p. 10)

We have expanded this section and added additional references to clarify the distinction and empirical support for dimensional approaches.

Dimensional models outperforming (p. 10)

The relevant paragraph has been revised to avoid overly strong claims and to provide clearer references supporting the statement.

Second-order categories (p. 11–13)

We have expanded the explanation of the second-order dimensions (sublime, uneasy, vital), including conceptual definitions, operationalisation in the GEMS framework and clarification that participants received standardised descriptors. We also included the intercorrelations among the first- and second-order factors in GEMS.

We also discuss potential interpretational variability in terms such as “sublime.”

“Valence” correction

Corrected throughout.

Strong claim (p. 12)

The statement has been moderated and additional references added.

Reductionism and isolation of parameters (p. 13)

We have strengthened the rationale for isolating tempo and mode and explicitly acknowledge the limitations of parameter-level approaches.

Submitted references

Removed or replaced as noted above.

Synoptic table and effect sizes (p. 17)

In response to this important suggestion, we have added a synoptic summary table (Table 1) presenting main effects and key interactions across all three emotional dimensions. Reported F-values, degrees of freedom, p-values, and partial eta squared consistently. Explicitly stated effect size benchmarks in the Analysis section. Clarified interpretation of Wilks’ lambda. Clarified how mean values were computed.

Full statistical tables are retained in supplementary materials for transparency.

Text in Discussion is better suited for Introduction (p. 24)

We have relocated the indicated paragraphs to the Introduction to improve structural coherence.

Figures (pp. 32 ff.)

All figures have been revised to include clearer captions and expanded explanatory text in the Results section to guide interpretation.

Concluding Remark

We are grateful for the reviewer’s detailed critique. The manuscript has undergone substantial revision to improve clarity, methodological transparency, theoretical positioning, and presentation of results. While the empirical findings remain unchanged, their structure and interpretation are now presented more clearly and concisely.

We hope the revised manuscript addresses the reviewer’s concerns and demonstrates the contribution of the study more convincingly.

Reviewer #3

Thank you for reviewing our manuscript. We appreciate the hard work and your comments.

1. Rephrase or substantiate the statement that tempo and mode are “inconsistently examined across emotion models.” Either specify the precise inconsistencies the paper addresses (e.g., differing stimuli, measures, or theoretical mappings) or soften the claim to position the work as a large-scale, stimulus-controlled extension of established findings. The manuscript itself cites many prior tempo/mode studies.

Thank you for this suggestion. We have rephrased this statement to avoid overgeneralization. Rather than implying broad inconsistency, we now position the study as a large-scale, stimulus-controlled extension of established findings, while acknowledging the substantial body of prior research on tempo and mode.

2) In the introduction, I recommend streamlining the sections on emotion models to improve logic: briefly introduce discrete, dimensional, and domain-specific approaches, then clearly justify the use of GEMS-9. The current text reads as somewhat disjointed.

We appreciate this comment. The previous version of the introduction had expanded in response to earlier reviewer requests for additional theoretical detail, which may have affected its coherence. We have now substantially streamlined the section by briefly introducing discrete, dimensional, and domain-specific approaches in a more structured sequence, followed by a clearer and more focused justification for the use of GEMS-9.

3) More importantly, flesh out the motivation for using GEMS second-order factors. Provide a stronger rationale for collapsing GEMS-9 to three second-order factors and report psychometric information, where available (internal reliabilities, inter-factor correlations…).

We agree that the rationale required further clarification. We have expanded the discussion of why the GEMS-9 second-order factors were selected, including their theoretical relevance and suitability for capturing music-specific emotional responses in large-scale designs. We now also report inter-factor correlations to increase transparency regarding the psychometric properties of the scales in our sample.

4) The sentence on emotional differences “emerging accidentally during the 14th–16th centuries” is ambiguous or unclear. Reword for clarity and cite an appropriate historical or psychoacoustic source (e.g., Parncutt, 2024).

Thank you for pointing this out. The sentence has been reworded for clarity, and we have added an appropriate reference to support the historical and psychoacoustic context.

5) Replication vs. extension – The study departs from previous methods and thus is not a strict replication. Please revise phrasing to “conceptual replication” or “extension,” and clearly distinguish replicated versus novel aspects.

We agree with this distinction. The study should indeed be described as a conceptual replication rather than a strict methodological replication. We have revised the terminology throughout the manuscript to clearly distinguish between replicated and novel aspects of the design.

6) Unclear terminology: “pure major” and “pure minor”

Thank you for noting this ambiguity. The terms “pure major” and “pure minor” have now been removed for clarity. The intention was simply to indicate that the stimuli were constructed in major and minor modes without modal mixture. As this terminology was potentially confusing and not essential, we have simplified the wording accordingly.

7) I understand the rationale behind using control stimuli, but couldn’t one raise the issue that control stimuli are auditory as well?

This is an important point. While alternative control modalities (e.g., visual stimuli) could have been used, we chose auditory control stimuli to ensure modality consistency and to isolate musical structure rather than sensory modality effects. We have now clarified this rationale in the introduction and explicitly addressed the limitations of this choice.

8) Describe the translation procedure (forward/back translation, expert review) and report internal consistency statistics.

We conducted forward and back translation procedures to ensure linguistic accuracy. Although an external expert review was not performed at the time, the translated instrument has since been used in multiple studies without issues. Unfortunately, internal consistency statistics were not computed in the original dataset; we acknowledge this as a limitation and clarify the translation procedure more explicitly in the manuscript.

9) Specify the hosting platform (e.g., Qualtrics, Gorilla, PsychoPy online), playback format, use of headphones...

We have added information about the hosting platform and relevant technical details in the Materials section. We previously mentioned headphone use in the limitations section; this has now also been clarified in the Procedure section for greater transparency.

---

## [Decision Letter · Decision Letter 2]

13 Apr 2026

The Effect of Tempi and Mode on the Rating of the Perceived Emotion in Music.

PONE-D-25-05889R2

Dear Dr. Færøvik,

We’re pleased to inform you that your manuscript has been judged scientifically suitable for publication and will be formally accepted for publication once it meets all outstanding technical requirements.

Kind regards,

Giulia Prete

Academic Editor

PLOS One

Additional Editor Comments (optional):

As you can see, one of the previous reviewers has agreed to review the new version of the manuscript and supports its publication in its current form. I have carefully read the revised version myself and agree that you have satisfactorily addressed the suggestions received. Therefore, I am happy to approve your manuscript for publication without further revisions.

Reviewers' comments:

Reviewer's Responses to Questions

**Comments to the Author**

1. If the authors have adequately addressed your comments raised in a previous round of review and you feel that this manuscript is now acceptable for publication, you may indicate that here to bypass the “Comments to the Author” section, enter your conflict of interest statement in the “Confidential to Editor” section, and submit your "Accept" recommendation.

Reviewer #2: All comments have been addressed

2. Is the manuscript technically sound, and do the data support the conclusions?

Reviewer #2: Yes

3. Has the statistical analysis been performed appropriately and rigorously? 

Reviewer #2: Yes

4. Have the authors made all data underlying the findings in their manuscript fully available?

Reviewer #2: Yes

5. Is the manuscript presented in an intelligible fashion and written in standard English?

Reviewer #2: Yes

6. Review Comments to the Author

Reviewer #2: (No Response)

7. PLOS authors have the option to publish the peer review history of their article (what does this mean?). If published, this will include your full peer review and any attached files.

Reviewer #2: No

---

## [Editor Report · Acceptance letter]

PONE-D-25-05889R2

PLOS One

Dear Dr. Færøvik,

I'm pleased to inform you that your manuscript has been deemed suitable for publication in PLOS One. Congratulations! Your manuscript is now being handed over to our production team.

Kind regards,

on behalf of

Dr. Giulia Prete

Academic Editor

PLOS One